# The Clinical Impact of the Omicron Variant on Octogenarian Hospitalized COVID-19 Patients: The Results from CoviCamp Cohort

**DOI:** 10.3390/biomedicines13071563

**Published:** 2025-06-26

**Authors:** Pierantonio Grimaldi, Mariantonietta Pisaturo, Antonio Russo, Salvatore Martini, Francesca Ambrisi, Filomena Milite, Giovanni Di Caprio, Fabio Giuliano Numis, Ivan Gentile, Vincenzo Sangiovanni, Vincenzo Esposito, Rossella Pacilio, Giosuele Calabria, Raffaella Pisapia, Canio Carriero, Alfonso Masullo, Elio Manzillo, Grazia Russo, Roberto Parrella, Sebastiano Leone, Michele Gambardella, Antonio Ponticiello, Nicola Coppola

**Affiliations:** 1Department of Mental Health and Public Medicine, University of Campania “L. Vanvitelli”, 80131 Naples, Italy; peogrimaldi@icloud.com (P.G.); mariantonietta.pisaturo@unicampania.it (M.P.); antonio.russo2@unicampania.it (A.R.); francescambrisi@gmail.com (F.A.); menamilite@yahoo.it (F.M.); 2Infectious Disease Unit, A.O.U. Luigi Vanvitelli, 80138 Naples, Italy; salvatoremartini76@gmail.com; 3Infectious Disease Unit, A.O.S Anna e S Sebastiano Caserta, 81100 Caserta, Italy; 4Emergency Unit, P.O. Santa Maria delle Grazie, 80078 Pozzuoli, Italy; fabiogiuliano.numis@aslnapoli2nord.it; 5Infectious Disease Unit, University Federico II, 80125 Naples, Italy; ivan.gentile@unina.it; 6Third Infectious Disease Unit, P.O. Cotugno, A.O.R.N. dei Colli, 80131 Naples, Italy; sangio.vincenzo@gmail.com; 7Infectious Diseases Unit and Gender Medicine, P.O. Cotugno, AORN dei Colli, 80131 Naples, Italy; vincenzoesposito@ospedalideicolli.it; 8Hepatic Infectious Disease Unit, P.O. Cotugno, A.O.R.N. dei Colli, 80131 Naples, Italy; rossella.pacilio@ospedalideicolli.it; 9IX Infectious Disease Unit, P.O. Cotugno, A.O.R.N. dei Coli, 80131 Naples, Italy; g.calabria@tin.it; 10First Infectious Disease Unit, P.O. Cotugno, A.O.R.N. dei Coli, 80131 Naples, Italy; raffipisapia@gmail.com; 11Infectious Disease Unit, P.O. Rummo, A.O. San Pio, 82100 Benevento, Italy; canio.carriero@aornsanpio.it; 12Infectious Disease Unit, A.O. San Giovanni di Dio e Ruggi D’Aragona Salerno, 84131 Salerno, Italy; alfonso.masullo@sangiovannieruggi.it; 13VIII Infectious Disease Unit, P.O. Cotugno, A.O.R.N. dei Coli, 80131 Naples, Italy; elio.manzillo@ospedalideicolli.it; 14Infectious Disease Unit, Ospedale Maria S.S. Addolorata di Eboli, A.S.L. Salerno, 84131 Salerno, Italy; gr.russo@aslsalerno.it; 15Respiratory Infectious Disease Unit, P.O. Cotugno, A.O.R.N. dei Colli, 80131 Naples, Italy; roberto.parrella@ospedalideicolli.it; 16Infectious Disease Unit, A.O. Avellino, 83100 Avellino, Italy; sebastianoleone@yahoo.it; 17Infectious Disease Unit, P.O. S. Luca, Vallo della Lucania, A.S.L. Salerno, 84131 Salerno, Italy; gambardella1960@gmail.com; 18Pneumology Unit, A.O.R.N. Caserta, 81100 Caserta, Italy; antonio.ponticiello@aorncaserta.it

**Keywords:** aging, omicron variant, COVID-19, SARS-CoV-2 infection

## Abstract

**Introduction:** This study aims to investigate the clinical impact of Omicron Severe Acute Respiratory Syndrome-Coronavirus-2 (SARS-CoV-2) infection on the clinical presentation of Coronavirus Disease 2019 (COVID-19) in the very old (≥80 years old) population. **Methods:** All patients aged 80 years or older, hospitalized from March 2020 to June 2023 with a SARS-CoV-2 infection in one of the 17 COVID-19 units in eight cities of Campania, southern Italy, were enrolled in a multicenter, observational, retrospective study. **Results:** 341 patients ≥ 80 years of age were included: 80 of them in the Omicron and 261 in the non-Omicron period. Patients admitted during the Omicron period were older (*p* = 0.0001) and more comorbid, showing more frequently arterial hypertension (*p* = 0.018), cardiovascular disease (*p* = 0.0001), chronic kidney disease (CKD) (*p* = 0.002), chronic obstructive pulmonary disease (COPD) (*p* = 0.001), and active cancer (*p* = 0.0001). Severe and critical outcomes were observed more often in the non-Omicron variant (*p* = 0.0001). Patients in the Omicron group did not show a significantly prolonged hospitalization time (*p* = 0.063) or a higher likelihood of death during hospitalization (*p* = 0.097). **Discussion:** In our study, despite the greater frailty of patients hospitalized during the Omicron period, the disease appeared less severe compared to previous waves, suggesting that the lower severity of the disease could be attributed to virological rather than population characteristics. These findings underscore the importance of prevention strategies for older people, as the administration of vaccination and early antiviral therapies in at-risk subjects.

## 1. Introduction

Age is commonly identified as the most significant risk factor for severe COVID-19 and its adverse health outcomes. Data from the early pandemic demonstrated that the case fatality ratio (CFR) of COVID-19 increases with age, reaching 20.2% in Italian patients aged 80 years, compared to a general CFR of 7.2% in the same setting [1]. During the first pandemic waves, the impact on older people has been widely investigated in terms of impact on COVID-19 outcome, constituting one of the concomitant conditions associated with an increased risk of negative outcome [2]. The risk associated with COVID-19 seems to be progressive along with age, resulting to require the recognition of a specific population of very old patients (≥80 years of age), with its specific characteristics: during the first pandemic waves, extreme old age itself was investigated as a possible condition influencing COVID-19 outcome, showing really high case fatality rates [3,4].

Since the end of 2021, with the appearance on scene of Omicron variant and its subvariants, clinical characteristics of COVID-19 have completely changed [5]; Omicron variant presents about 15 mutations in the Receptor Binding Domain region of the Spike protein, compared to the only 4 of the Delta variant: this element immediately rose attention for the higher spread risk. Under the clinical aspect, Omicron presented with milder, upper respiratory tract symptoms, with few patients presenting Computed Tomography lung infiltrates [6].

Data regarding the impact of very old age on clinical presentation of COVID-19 in the Omicron variant era are lacking; this study aims to evaluate the clinical impact of the Omicron variant in the very old population.

## 2. Materials and Methods

### 2.1. Study Design and Setting

We performed a multicentre, observational, retrospective study involving seventeen COVID-19 units in eight cities in the Campania region in southern Italy: Naples, Caserta, Salerno, Benevento, Avellino, Pozzuoli, Eboli, and Vallo della Lucania. All adult (≥18 years) patients, hospitalized with a diagnosis of SARS-CoV-2 infection, confirmed by RT-PCR on a naso-oropharyngeal swab from 28 February 2020 to 30 June 2023 at one of the centres participating in the study, were enrolled in the CoviCamp cohort [7,8,9,10,11,12]. No study protocol or guidelines regarding the criteria of hospitalization were shared among the centres involved in the study, and the patients were hospitalized following the decision of the physicians of each center.

From the CoviCamp cohort, for the present analysis, we included all patients who were more than 65 years old. Exclusion criteria included lack of clinical data and/or informed consent.

The study was approved by the Ethics Committee of the University of Campania L. Vanvitelli, Naples (n°10877/2020). All procedures performed in this study were in accordance with the ethics standards of the institutional and/or national research committee and with the 1964 Helsinki declaration and its later amendments or comparable ethics standards. Informed consent was obtained from all participants included in the study.

This study was reported following the STROBE recommendations for an observational study (Appendix A).

### 2.2. Variables and Definitions

All demographic and clinical data of patients with SARS-CoV-2 infection enrolled in the cohort were collected in an electronic database. From this database, we extrapolated the data for the present study.

The microbiological diagnosis of SARS-CoV-2 infection was defined as a positive RT-PCR test on a naso-oropharyngeal swab. All of the units included used the same RT-PCR kit, Bosphore V3 (Anatolia Genework, Sultanbeyli, Turkey). We considered the time-to-negative swab as the days from the first positive RT-PCR for SARS-CoV-2 to the first negative RT-PCR; the data were extrapolated from the regional database (https://sinfonia.regione.campania.it/preview/ecovid, last accessed on 1 December 2023). Where available, we included the time from the last positive nasopharyngeal swab to the first negative nasopharyngeal swab for SARS-CoV-2. According to Italian official data [13] from 3 January 2022, in Italy, 80.4% of the variants isolated were Omicron. Therefore, we decided to include all patients with a diagnosis of SARS-CoV-2 infection from January 3th 2022 into the Omicron variant group.

We divided the patients enrolled according to the clinical outcome of COVID-19: patients with a mild outcome were those who did not require oxygen therapy, those who performed oxygen therapy with a nasal cannula or venturi mask, hemodynamically stable patients, and patients with a Glasgow Coma Scale greater than 9; patients with severe outcome were those who underwent oxygen therapy with high flow nasal cannula or non-invasive or invasive ventilation; non-hemodynamically stable patients were those with a Glasgow Coma Scale less than 9.

Exclusively for the outcome mortality, we conducted a power analysis to determine whether to detect any risk of underestimating a difference between groups due to sample size. We also performed a multivariate analysis on factors associated with mortality of the ≥80 years of age (y/o) patients.

### 2.3. Statistical Analysis

For the descriptive analysis, categorical variables were presented as absolute numbers and their relative frequencies. Continuous variables were summarized as mean and standard deviation or as median and interquartile range (Q1–Q3). We performed a comparison of patients hospitalized during Omicron waves and during non-Omicron waves using Pearson’s chi-square or Fisher’s exact test for categorical variables and Student’s t-test or Mann–Whitney test for continuous variables. Power analysis was performed with power analysis for independent proportions. Multivariate analysis was performed through multiple logistic regression. Analyses were performed by STATA 16 [14] and SPSS Statistics 19 [15].

## 3. Results

Among the 2083 patients admitted to one COVID-19 unit included in the study, 1090 were excluded because they were under 65 years of age. Of the 993 patients included in the study, 652 were between 65 and 79 years of age (y/o), while the remaining 341 were at least 80 years old. Among 341 patients aged 80 years or older, 80 belonged to the Omicron variant wave, and 261 were from different waves (Figure 1). A comparison of patients between 65 and 80 y/o versus patients with at least ≥80 y/o was performed (Appendix A). As shown in Table 1, compared to the 261 octogenarian patients in non-Omicron period, the 80 patients admitted in hospital during Omicron wave were older (91 y/o vs. 83 y/o, *p* = 0.0001), had more frequently arterial hypertension (63.3% vs. 48%, *p* = 0.018), cardiovascular disease (48.1% vs. 25.7%, *p* = 0.0001), CKD (21.1% vs. 8.3%, *p* = 0.002), COPD (36.8% vs. 12.3%, *p* = 0.001), and active cancer (35.1% vs. 7.5%, *p* = 0.0001) (Table 1). Patients in the non-Omicron group were more frequently symptomatic compared to patients included in the Omicron group (Table 1). Specifically, in the non-Omicron group, patients showed more frequent fever (57.3% vs. 28%, *p* = 0.001), cough (29.8% vs. 15.3%, *p* = 0.015), and dyspnea (67% vs. 50%, *p* = 0.010) (Table 1). Considering biochemical data, the octogenarian patients in the Omicron group showed a higher PaO2/FiO2 ratio (*p*/F) compared to the non-Omicron group (median 247 vs. 214, *p* = 0.044) (Table 1), as well as lower leukocyte and neutrophil count, and higher lymphocyte count (Table 2). More severe and critical outcomes were observed in the non-Omicron variant (36.3% vs. 12.5%, *p* = 0.0001) (Table 1). Patients with the Omicron group showed no significantly prolonged hospitalization compared with the non-Omicron group (median 12 days vs. 14 days, *p* = 0.063), as well as death during hospitalization (5.6% vs. 12.7%, *p* = 0.097) (Table 1). For the outcome mortality, the power analysis attested a power of 0.410 to detect a difference in mortality between the Omicron era and non-Omicron era patients. Comparing patient morbidity during Omicron and non-Omicron waves, analysis showed that patients in Omicron waves were older than the comparator group (90 y/o vs. 82 y/o, *p* < 0.0001), and patients hospitalized showed a lower rate of continuous positive airways pressure or non-invasive ventilation (CPAP/NIV) use: 50% in non-Omicron waves, and 0% in Omicron wave (Table 3). Factors associated with mortality are shown in Appendix A.

## 4. Discussion

This study analyzes the clinical and prognostic differences between elderly patients hospitalized for COVID-19 during the Omicron wave compared to those infected with previous variants. With the possible bias due to lack of shared protocols to determine hospitalization, the results highlight how patients hospitalized during the Omicron wave were significantly older (91 vs. 83 years, *p* = 0.0001) and with a greater burden of comorbidities, including a high prevalence of arterial hypertension, cardiovascular disease, CKD, COPD, and active cancers. However, despite this greater frailty, the pulmonary disease was less severe compared to previous waves. These data are consistent with what has been reported by several studies that have shown a reduced virulence of the Omicron variant compared to previous variants, such as Delta. Nyberg et al. [16] demonstrated that the adjusted hazard ratio (HR) estimates for hospital admission and death with Omicron compared with Delta were 0.41 (0.39–0.43) and 0.31 (0.26–0.37), respectively.

Furthermore, Omicron versus Delta HR estimates varied with age; adjusted HR for hospital admission was 1.10 (0.85–1.42) in those younger than 10 years, decreasing to 0.25 (0.21–0.30) in 60–69-year-olds, and then increasing to 0.47 (0.40–0.56) in those aged at least 80 years [16].

Additionally, another study conducted in the US found that hospital mortality among patients hospitalized primarily for COVID-19 decreased from 15.1% during the Delta variant period to 4.9% in the subsequent period with the Omicron variant, despite a higher proportion of high-risk patients among those hospitalized [17].

Our study showed a different clinical presentation between COVID-19 patients during the Omicron era and those of the previous periods. Omicron wave patients presented less frequently with symptoms such as fever (28% vs. 57.3%, *p* = 0.001), cough (15.3% vs. 29.8%, *p* = 0.015), and dyspnea (50% vs. 67%, *p* = 0.010), suggesting a less pronounced systemic involvement. Furthermore, the PaO2/FiO2 (P/F) ratio was significantly higher in Omicron era patients (247 vs. 214, *p* = 0.044), indicating a lower degree of respiratory failure. The percentage of severe and critical outcomes was significantly lower in Omicron wave patients (12.5% vs. 36.3%, *p* = 0.0001), confirming what has already been observed in other studies (1,2). Our patients did not undergo any genomic testing to confirm the variant they were affected by, and our considerations on the Omicron variant are based on a certainly strong, but still purely epidemiological, basis [13]. With this limitation, our results support the hypothesis that the Omicron variant has a greater predilection for the upper respiratory tract rather than the lung parenchyma, thus reducing the frequency of severe interstitial pneumonia and the consequent need for invasive mechanical ventilation. Experimental data have shown that Omicron has a lower capacity to infect alveolar cells compared to Delta, which could explain the lower incidence of severe respiratory failure observed in our patients [18,19,20].

Another relevant aspect is the different use of therapeutic strategies and the introduction and diffusion of vaccines between the different waves. During the first phase of the pandemic, the therapeutic approach was more aggressive, with a large use of high-dose corticosteroids, anticoagulants, and non-invasive ventilation (NIV). In our study, a reduced need for NIV and CPAP was observed in Omicron era patients compared to non-Omicron era patients (0% vs. 50%), suggesting a switch in the clinical phenotype of the in-patient in the Omicron era, more comorbid and less compromised at an acute respiratory level. However, it is necessary to underline that in this cohort, the data on antiviral use were not available.

An additional point of interest concerns in-hospital mortality. Although the percentage of deaths was numerically lower in the Omicron era group compared to the non-Omicron era group (5.6% vs. 12.7%, *p* = 0.097), this difference did not reach statistical significance. Seemingly, the higher a priori complexity of in-patients of the latter pandemic waves did not result in higher mortality: this can be due to the lower virulence of the variant; a further element to consider is that the attested power for the analysis involving this outcome resulted in 0.410, with possible underestimation. Other studies have shown that, despite the reduced severity of the disease, mortality in elderly patients remains high due to intrinsic frailty and pre-existing comorbidities. For example, a systematic review examined the clinical outcomes of hospitalized patients over 65 with the Omicron variant. Despite the overall lower severity of the variant, older adults remain at risk of developing severe symptoms [19]. Another study found that during the Omicron variant predominance period, overall in-hospital mortality decreased compared to the Delta variant period. However, the majority of in-hospital deaths occurred among adults aged ≥65 years, underlining the persistent vulnerability of this age group [16].

These studies indicate that, although the Omicron variant may cause less severe symptoms in the general population, individuals over 80 years of age remain at high risk of severe complications. Furthermore, our study confirms the importance of age as the main negative prognostic factor in COVID-19. As previously reported by studies conducted in China, Germany, Italy, South Korea, Spain, the United States, and New York City, the case fatality rate increased dramatically with age [21,22]. Although less virulent, the Omicron variants seem to retain the ability to cause severe disease in older patients. Therefore, it is essential to continue implementing preventive measures, such as vaccination and the use of antiviral treatments, to protect this vulnerable population.

## 5. Conclusions

Our study confirms that the Omicron variant had a less severe impact than previous variants in elderly patients hospitalized for COVID-19. However, patients hospitalized during the Omicron wave were older and had a higher comorbidity burden, suggesting that the lower severity of the disease could be attributed to virological characteristics rather than a less vulnerable population.

These findings underscore the importance of targeted prevention strategies for older people, including vaccination and early administration of antiviral therapies in at-risk subjects. Furthermore, they highlight the need for further studies to better understand the mechanisms by which Omicron leads to less severe disease, despite increased transmissibility.

## Figures and Tables

**Figure 1 biomedicines-13-01563-f001:**
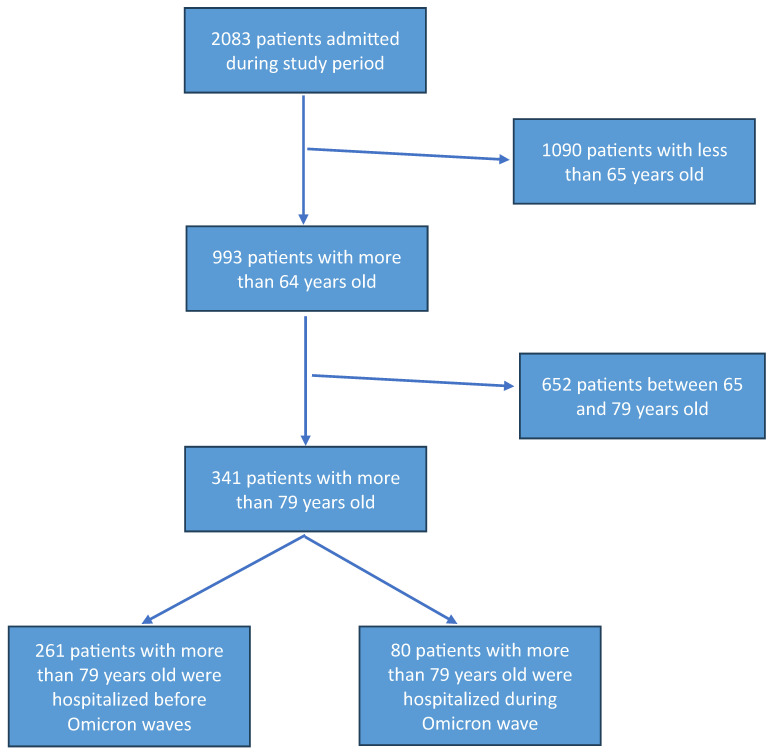
Flow chart of patients included in the study.

**Table 1 biomedicines-13-01563-t001:** Demographic and clinical characteristics of patients at least 80 years old grouped by the SARS-CoV-2 variant.

	At Least 80 y/o Patients During Other Waves	At Least 80 y/o Patients in Omicron Era	*p* Value
**Number of patients**, *n* (%)	261	80	-
**Age**, median [IQR]	83 [81–85]	91 [88–92]	**0.0001**
**Males**, *n* (%)	128 (49)	33 (41.3)	0.222
**Charlson Comorbidity Index**, median [IQR]	2 (1–4)	5 (4–7)	**0.0001**
**Hypertension**, *n* (%)	120 (48)	50 (63.3)	**0.018**
**Diabetes**, *n* (%)	60 (23.8)	22 (28.8)	0.432
**Overweight/Obesity**, *n* (%)	15 (9.4)	2 (8.7)	1
**COPD**, *n* (%)	31 (12.3)	28 (36.8)	**0.0001**
**Cardiovascular pathology**, *n* (%)	65 (25.7)	37 (48.1)	**0.0001**
**Dementia**, *n* (%)	15 (7.5)	9 (12.2)	0.221
**HIV**, *n* (%)	2 (0.8)	0 (0)	1
**Active tumor**, *n* (%)	19 (7.5)	27 (35.1)	**0.0001**
**Chronic liver disease**, *n* (%)	14 (5.7)	2 (2.6)	0.278
**CKD**, *n* (%)	21 (8.3)	16 (21.1)	
**Fever**, *n* (%)	125 (57.3)	21 (28)	**0.0001**
**Cough**, *n* (%)	65 (29.8)	11 (15.3)	**0.015**
**Asthenia**, *n* (%)	36 (17.8)	20 (27.4)	0.082
**Ageusia/Dysgeusia**, *n* (%)	7 (3.2)	0 (0)	0.196
**Anosmia/hyposmia**, *n* (%)	7 (3.2)	0 (0)	0.351
**Diarrhea**, *n* (%)	8 (3.7)	6 (7.9)	0.136
**Skin lesions**, *n* (%)	1 (0.5)	6 (8.7)	**0.001**
**Dyspnea**, *n* (%)	146 (67)	36 (50)	**0.010**
**P/F**, median [IQR]	214 [137–314]	247 [202–319]	**0.044**
**Time-to-negative swab**, median [IQR]	14 [0–21]	14 [9–17]	0.494
**Hospitalization days**, median [IQR]	14 [9–19]	12 [9–17]	0.054
**Severe/critical outcome**, *n* (%)	77 (36.3)	9 (12.5)	**0.0001**
**Death**, *n* (%)	27 (12.7)	4 (5.6)	0.124

Abbreviations: COPD: chronic obstructive pulmonary disease; HIV: Human Immunodeficiency Virus; CKD: chronic kidney disease; P/F: partial pressure of O_2_/inspired fraction of O_2_. In bold, *p*-values < 0.05.

**Table 2 biomedicines-13-01563-t002:** Hematobiochemical presentation of patients at least 80 years of age, grouped by SARS-CoV-2 variant.

	Other Waves, at Least 80	Omicron Era, at Least 80	*p* Value
**WBC**, median [IQR]	7970 [5800–10,300]	6180 [4540–8100]	**0.0001**
**Lymphocytes**, median [IQR]	880 [640–1251]	1127 [659–1742]	**0.009**
**Neu**, median [IQR]	6380 [4720–8620]	3964 [2459–5754]	**0.0001**
**INR**, median [IQR]	1.12 [1.04–1.24]	1.07 [1.0–1.2]	**0.041**
**Creatinine**, median [IQR]	0.9 [0.7–1.10]	0.87 [0.71–1.12]	0.741
**CPK**, median [IQR]	74 [42–146]	44 [26–70]	**0.0001**
**GPT**, median [IQR]	28 [20–51]	23 [10–38]	**0.008**
**Direct bilirubinemia**, median [IQR]	0.22 [0.17–0.35]	0.29 [0.20–0.45]	0.800

Abbreviations: WBC: White Blood Cells; Neu: Neutrophils; INR: International Normalized Ratio; CPK: creatinin-phospho-kinase; GPT: Glutamate Pyruvate Transaminase. In bold, *p*-values < 0.05

**Table 3 biomedicines-13-01563-t003:** Demographic and clinical characteristics of patients over 80 years old with in-hospital mortality, grouped by SARS-CoV-2 variant.

Patient Mortality During Hospitalization	Other Waves, at Least 80	Omicron Era, at Least 80	*p* Value
**Number of patients**, *n* (%)	27 (12.7)	4 (5.6)	-
**Age**, median [IQR]	82 [81–85]	90 [88–96]	**0.0001**
**Males**, *n* (%)	12 (44.4)	3 (75)	0.333
**Charlson Comorbidity Index**, median [IQR]	6 (5–9)	9 (7–10)	0.181
**Hypertension**, *n* (%)	14 (53.8)	3 (75)	0.613
**Diabetes**, *n* (%)	12 (46.2)	0 (0)	0.130
**Overweight/Obesity**, *n* (%)	2 (15.4)	0 (0)	1
**COPD**, *n* (%)	5 (19.2)	2 (50)	0.225
**Cardiovascular pathology**, *n* (%)	12 (46.2)	3 (75)	0.598
**Dementia**, *n* (%)	7 (38.9)	3 (75)	0.93
**HIV**, *n* (%)	0 (0)	0 (0)	-
**Active tumor**, *n* (%)	7 (26.9)	2 (50)	0.563
**Chronic liver disease**, *n* (%)	2 (7.7)	0 (0)	1
**CKD**, *n* (%)	6 (23.1)	2 (50)	0.284
**Fever**, *n* (%)	7 (35)	0 (0)	0.526
**Cough**, *n* (%)	4 (20)	0 (0)	1
**Asthenia**, *n* (%)	4 (21.1)	2 (66.7)	0.169
**Ageusia/Dysgeusia**, *n* (%)	0 (0)	0 (0)	-
**Anosmia/hyposmia**, *n* (%)	0 (0)	0 (0)	-
**Diarrhea**, *n* (%)	0 (0)	0 (0)	-
**Skin lesions**, *n* (%)	1 (5)	0 (0)	1
**Dyspnea**, *n* (%)	12 (60)	2 (66.7)	1
**Hospitalization days**, median [IQR]	8 [4–12]	2 [1–21]	0.464
**Oxygen administration mode**, *n* (%)			
***Nasal cannulas/Venturi Mask***	9 (50)	3 (75)	
***High-flow nasal cannulas***	0 (0)	1 (25)	**0.032**
***CPAP/NIV***	9 (50)	0 (0)	

Abbreviations: COPD: chronic obstructive pulmonary disease; HIV: Human Immunodeficiency Virus; CKD: chronic kidney disease; CPAP: Continuous Positive Airway Pressure; NIV: non-invasive ventilation. In bold, *p*-values < 0.05

## Data Availability

Data is contained within the article/Appendix A.

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
