# Peer review of "The Clinical Impact of the Omicron Variant on Octogenarian Hospitalized COVID-19 Patients: The Results from CoviCamp Cohort"

_biomedicines, 2025, doi:10.3390/biomedicines13071563_

Round 1
Reviewer 1 Report
Comments and Suggestions for Authors
This study aims to investigate the clinical impact of Omicron SARS-CoV-2 infection on clinical presentation of COVID-19 in the very old population. Unfortunately, I have a few comments:
- A lot of typos in the text
- Why the different ages of the study participants are given (>80 years old, over 79 years, over 80 years)?
- Was there a confirmed SARS-CoV-2 variant in the patients? If not, how do you know that participants in the 'omicron era' had an omicron variant?
Author Response
Dear Editor,
We re-submit our paper “THE CLINICAL IMPACT OF THE OMICRON VARIANT ON OCTOGENARIAN HOSPITALIZED COVID-19 PATIENTS: THE RESULTS FORM COVICAMP COHORT.” (manuscript n°: biomedicines-3635461), modified according to the suggestions of the Reviewers.
First, we would like to thank you all for the time you’ve taken to read and evaluate our work. We sincerely think the comments and suggestion you gave us will improve our paper; we have thoroughly considered each of those points and we present below our point-by-point responses to each of the reviewers’ comments; attached, you’ll find the manuscript with the necessary improvements.
ANSWERS TO THE COMMENTS OF THE REVIEWER 1
Point 1. A lot of typos in the text
Thank you, we corrected these typos in the current version of the paper.
Point 2. Why the different ages of the study participants are given (>80 years old, over 79 years, over 80 years)?
As required by the reviewer, we specified that the real cut off was being at least 80 years of age.
Point 3. Was there a confirmed SARS-CoV-2 variant in the patients? If not, how do you know that participants in the 'omicron era' had an omicron variant?
Thanks for the question. We did not routinely perform any test to detect specific COVID-19 variants in-hospital patients, so we do not have any certainty of the variant our patients were affected with. Anyway, Omicron variant had a significant burst at the end of 2021, with a report, cited in our bibliography at the voice 12, stating that in Italy already at January 3th, 2022 80.4% of the isolated circulating variant were omicron. Attempting to avoid ambiguity, in the text and tables we now increased the references to “omicron era”, due to lack of certainty and not to “patients affected from omicron variant”. We also added a sentence to the Discussions to further address this issue.
ANSWERS TO THE COMMENTS OF THE REVIEWER 2
Point 1. General Assessment
This manuscript presents a multicenter, retrospective study evaluating the clinical characteristics and outcomes of hospitalized COVID-19 patients over the age of 80 during the Omicron wave compared to previous waves in Southern Italy. The topic is timely and of clinical relevance, given the aging population and evolving SARS-CoV-2 variants. The study is well structured and generally clearly written. However, several issues related to methodology, data interpretation, and clarity should be addressed before the manuscript is considered for publication.
Thank you very much. We are glad you our study, despite some issues, interested you. We will try to improve our work according to your suggestion in order to at least mitigate some of your legittimate concerns.
Point 2. Definition of Omicron Period Without Genomic Confirmation
The classification of Omicron cases based solely on the calendar date (from January 3, 2022) without genomic confirmation is a major limitation. It introduces the possibility of misclassification, especially during the transition between variant waves. The authors should clearly discuss this limitation and, if available, provide any genomic surveillance data to support the temporal threshold used.
Thank you for this comment. We specified, in order to reduce ambiguity, anytime in the text and tables, that patients were selected on an epidemiological (e.g: Omicron wave, era) rather than genomical base. Moreover, we’ve added a sentence to the Discussions to further specificy this issue.
Point 3. Lack of Treatment Data
The manuscript does not provide information on therapeutic interventions (e.g., antivirals, corticosteroids, oxygen therapy, CPAP/NIV use) that could confound the comparison of clinical outcomes. These data are critical to interpreting the observed differences in severity and mortality. Inclusion or discussion of these factors is recommended.
Unfortunately, data on corticosteroids and antiviral use were only available for a minority of patients. We reported in table 3 data on oxigen administration/ CPAP/NIV use. We added this limit in the discussion section
Point 4. Mortality Comparison and Statistical Power
Although the in-hospital mortality rate was numerically lower in the Omicron group, the difference did not reach statistical significance (p=0.097). The authors should clarify whether the study was powered to detect differences in mortality and consider discussing this limitation in the discussion section.
Thank you; we took your advice and performed a power analysis, reported in methods, results, and discussion section. The detected power was probably low (0.410) and we reported on that.
Point 5. Standardization of Severity Criteria Across Centers
The study relies on data from 17 centers without a standardized hospitalization or severity scoring system. Variability in clinical decision-making could affect the comparability of outcomes. This heterogeneity should be addressed as a source of potential bias.
Thank you for your suggestion. This surely constitutes a bias: we added a sentence in the section Discussions.
Point 6. Minor Comments
Clarity of Figures and Tables
Ensure that all tables are clearly labeled with complete legends. For example, Table 1 should explicitly define abbreviations such as P/F, COPD, and CKD.
Thank you, we now added footnotes.
Point 7. Language and Grammar
While generally well-written, the manuscript would benefit from professional language editing to improve readability and correct minor grammatical errors.
Thank you. A deep proof-reading has been performed, with relative corrections.
Point 8. Ethical Approval and Consent
The authors mention ethical approval and informed consent; however, it would be helpful to explicitly state that consent was obtained from legal representatives where appropriate, considering the advanced age of participants.
Thank you, we modified the manuscript accordingly.
ANSWERS TO THE COMMENTS OF THE REVIEWER 3
Point 1. Abstract and Introduction • Clarity of the Problem Statement: The article effectively addresses a highly relevant question by comparing the clinical impact of the Omicron variant versus previous variants in very elderly patients. The introduction is grounded in the existing literature by emphasizing the prognostic importance of age in COVID-19 outcomes. However, it would be beneficial to expand on how other factors, such as vaccination status, early antiviral treatment, and evolving therapeutic protocols have played a role as the pandemic has progressed. • Suggestion: Consider briefly discussing the evolution of public health strategies over the pandemic, as changes in vaccination coverage and clinical management might have influenced patient outcomes during the Omicron wave.
Thank you; we addressed these issues in the discussion section.
Point 2. Inclusion and Exclusion Criteria: The authors begin with patients over 65 years of age and then focus on those over 79. It would be helpful to elaborate on the rationale for choosing octogenarians as the primary group for analysis. Explaining why this particular age subgroup was selected would further justify the study focus and strengthen the manuscript.
Thank you for your comment: from our experience and literature study [see bibliography 3, 4] we had the feeling very old patients have different outcomes when affected with COVID 19 even when compared to old patients. For this reason, we carried out this observational study on subjects over 79 years old.
Point 3. Statistical Analysis: Suggestion; including a multivariate analysis (such as logistic regression) that adjusts for potential confounders—age, comorbidity burden, etc.—could enhance the validity of the findings by providing an independent assessment of the variant effect on disease severity. Results • Suggestions: o Discuss further how changes in treatment protocols (e.g., reduced use of CPAP/NIV in the Omicron period) might have contributed to these findings. o Address why the difference in in-hospital mortality did not reach statistical significance, which could be due to sample size or other confounders that were not controlled for.
Thank you: as suggested, we performed a multivariable analysis on factors associated with mortality, reported in supplementary material. We’ve also carried out a power analysis, showing a possible undersizing explainining the lack of significant difference in mortality.
Point 4. Confounding Factors: It would be beneficial to discuss more explicitly the limitations regarding the heterogeneity in patient management among different centers, variations in therapeutic strategies, and potential biases arising from classifying the variant based solely on the date of diagnosis rather than through specific viral sequencing for each patient.
Thank you. In the discussion section of the new manuscript we reported these limits.
Point 5. Vaccination and Antiviral Therapies: Since vaccination status and early therapies have likely influenced outcomes, acknowledging either the data available or the absence of such data as a limitation would add nuance to the discussion.
Thank you. We now specifically adressed these points in the discussion section.
Point 6. The tables and figures should have uniform formatting with complete legends to make them interpretable on their own.
Thank you. We’ve now integrated tables with appropriate footnotes.
Point 7. There should be this section and place the limitations that they found when doing the study.
Thank you. We now specifically addressed study limitations in the discussion section.
ANSWERS TO THE COMMENTS OF THE REVIEWER 4
Point 1. In the Abstract section, you are kindly invited to replace "Severe Acute Respiratory Distress Syndrome-Coronavirus-2" by "Severe Acute Respiratory Syndrome-Coronavirus-2".
Thank you for noticing this mistake. We modified accordingly.
Point 2. In the Article you talked about "Omicron Period" and "Non-Omicron Period" How can you define these two periods?
We started considering patients as belonging to the Omicron wave if they were admitted after january the 3rd, 2022, according to italian surveilance data [12]. We have now addressed the limitations of this method compared to a genomic investigation of the patients in the discussion section.
Point 3. In several places of the manuscript, when you want to use an abbreviation, you are kindly asked (for the first time) to put the full name followed by its abbreviation (between parenthesis) then you can use the abbreviation. Example in the Abstract "CKD" and "COPD".
We modified the text according to the suggestion of the reviewer.
Point 4. The Introduction section is very short, I invited you to talk more about SARS-CoV-2 variants, and the impact of this variation on the transmission and the virulence of the virus in humans. I suggest to use the following articles as references for this point:
Thank you for your suggestion. We expanded the section by mentioning a suggested work.
Point 5. In your tables, you are kindly invited the full words of your abbreviations below the tables.
Thank you; we added appropriate footnotes.
Point 6. In your hematological parameters, why you don't have inflammatory markers such as CRP or ESR?
Unfortunately, most centres did not provide the data, so we didn’t report it to avoid excessive heterogeneity.
We thank the Reviewers for helping us to improve our paper.
We hope that the paper is now worthy of publication in “Biomedicines”
Best regards,
Prof Nicola Coppola

Reviewer 2 Report
Comments and Suggestions for Authors
General Assessment
This manuscript presents a multicenter, retrospective study evaluating the clinical characteristics and outcomes of hospitalized COVID-19 patients over the age of 80 during the Omicron wave compared to previous waves in Southern Italy. The topic is timely and of clinical relevance, given the aging population and evolving SARS-CoV-2 variants. The study is well structured and generally clearly written. However, several issues related to methodology, data interpretation, and clarity should be addressed before the manuscript is considered for publication.
Major Comments
Definition of Omicron Period Without Genomic Confirmation
The classification of Omicron cases based solely on the calendar date (from January 3, 2022) without genomic confirmation is a major limitation. It introduces the possibility of misclassification, especially during the transition between variant waves. The authors should clearly discuss this limitation and, if available, provide any genomic surveillance data to support the temporal threshold used.
Lack of Treatment Data
The manuscript does not provide information on therapeutic interventions (e.g., antivirals, corticosteroids, oxygen therapy, CPAP/NIV use) that could confound the comparison of clinical outcomes. These data are critical to interpreting the observed differences in severity and mortality. Inclusion or discussion of these factors is recommended.
Mortality Comparison and Statistical Power
Although the in-hospital mortality rate was numerically lower in the Omicron group, the difference did not reach statistical significance (p=0.097). The authors should clarify whether the study was powered to detect differences in mortality and consider discussing this limitation in the discussion section.
Standardization of Severity Criteria Across Centers
The study relies on data from 17 centers without a standardized hospitalization or severity scoring system. Variability in clinical decision-making could affect the comparability of outcomes. This heterogeneity should be addressed as a source of potential bias.
Minor Comments
Clarity of Figures and Tables
Ensure that all tables are clearly labeled with complete legends. For example, Table 1 should explicitly define abbreviations such as P/F, COPD, and CKD.
Language and Grammar
While generally well-written, the manuscript would benefit from professional language editing to improve readability and correct minor grammatical errors.
Ethical Approval and Consent
The authors mention ethical approval and informed consent; however, it would be helpful to explicitly state that consent was obtained from legal representatives where appropriate, considering the advanced age of participants.
Conclusion
The manuscript addresses an important topic and provides valuable data on elderly COVID-19 patients during the Omicron era. However, several methodological concerns, particularly the absence of genomic confirmation and treatment data, limit the strength of the conclusions. With revisions and clarifications, this study could contribute meaningfully to the literature on COVID-19 outcomes in geriatric populations.
Author Response

(The authors gave the same response as above.)

Reviewer 3 Report
Comments and Suggestions for Authors
Thank you for submitting your manuscript. Please refer to the attached file for detailed suggestions and comments that we believe will help enhance the clarity and overall impact of your work.

Author Response

(The authors gave the same response as above.)

Reviewer 4 Report
Comments and Suggestions for Authors
Dear Authors,
Your article entitled "The clinical impact of the Omicron Variant on octogenarian Hospitalized COVID-19 Patients: the results form CoviCamp cohort" has been carefully reviewed,
The present article is important since it highlights on the clinical impact of one of the famous variants of SARS-CoV-2 the "omicron variants" on Elderly patients (aged more than 79 years old) in a region in the south of Italy.
The article is well written, well designed, the information are clear for readers, tables and figures are very simple and direct to the point.
Kindly find below the list of comments concerning this work:
01- In the Abstract section, you are kindly invited to replace "Severe Acute Respiratory Distress Syndrome-Coronavirus-2" by "Severe Acute Respiratory Syndrome-Coronavirus-2".
02- In the Article you talked about "Omicron Period" and "Non-Omicron Period" How can you define these two periods?
03- In several places of the manuscript, when you want to use an abbreviation, you are kindly asked (for the first time) to put the full name followed by its abbreviation (between parenthesis) then you can use the abbreviation. Example in the Abstract "CKD" and "COPD".
04- The Introduction section is very short, I invited you to talk more about SARS-CoV-2 variants, and the impact of this variation on the transmission and the virulence of the virus in humans. I suggest to use the following articles as references for this point:
-- The Emergence of SARS-CoV-2 Variant(s) and Its Impact on the Prevalence of COVID-19 Cases in the Nabatieh Region, Lebanon
-- A comparative overview of SARS-CoV-2 and its variants of concern
05- In your tables, you are kindly invited the full words of your abbreviations below the tables.
06- In your hematological parameters, why you don't have inflammatory markers such as CRP or ESR?
Best Regards,
Author Response

(The authors gave the same response as above.)

Round 2
Reviewer 4 Report
Comments and Suggestions for Authors
Dear Authors,
The revised version of your manuscript has been carefully reviewed,
The paper is more suitable for publication in its present form, thanks to the modifications you made,
Best Regards,